# Multi-population stochastic modeling of Ebola in Sierra Leone: Investigation of spatial heterogeneity

Rachid Muleia [1,2]*, Marc Aerts[1], Christel Faes[1]

**1** Interuniversity Institute for Statistics and Statistical Bioinformatics, Data Science Institute, Hasselt University, Diepenbeek, Belgium, **2** Department of Mathematics and Informatics, Faculty of Sciences, Eduardo Mondlane University, Maputo, Mozambique

* rmuleia@gmail.com

**Data Availability Statement:** The data used in this manuscript can be found at https://github.com/RachidMuleia/Sierra_Leone_Data.

**Funding:** The present study received financial support from Universidade Eduardo Mondlane

## Abstract

A major outbreak of the Ebola virus occurred in 2014 in Sierra Leone. We investigate the spatial heterogeneity of the outbreak among districts in Sierra Leone. The stochastic discrete-time susceptible-exposed-infectious-removed (SEIR) model is used, allowing for probabilistic movements from one compartment to another. Our model accounts for heterogeneity among districts by making use of a hierarchical approach. The transmission rates are considered time-varying. It is investigated whether or not incubation period, infectious period and transmission rates are different among districts. Estimation is done using the Bayesian formalism. The posterior estimates of the effective reproductive number were substantially different across the districts, with pronounced variability in districts with few cases of Ebola. The posterior estimates of the reproductive number at the district level varied between below 1.0 and 4.5, whereas at nationwide level it varied between below 1.0 and 2.5. The posterior estimate of the effective reproductive number reached a value below 1.0 around December. In some districts, the effective reproductive number pointed out for the persistence of the outbreak or for a likely resurgence of new cases of Ebola virus disease (EVD). The posterior estimates have shown to be highly sensitive to prior elicitation, mainly the incubation period and infectious period.

## Introduction

Ebola, also known as Ebola hemorrhagic fever, is a rare and deadly disease caused by an infection of one of the Ebola virus strains. Ebola can cause diseases in humans and nonhuman primates (monkeys, gorillas, and chimpanzees) [1]. EVD is mainly transmitted through direct contact with infected bodily fluids and contaminated materials. At the early stage, the disease is characterized by initial-flu symptoms, high fever, severe headache followed by pharyngitis and abdominal pain, whereas the late stage of the disease is marked by vomiting, diarrhea, rash, and internal and external bleeding [2]. The disease was first identified in Sudan and Zaire in 1976 where it infected over 284 people with a mortality rate of 53%, and subsequently, few

(UEM), Hasselt University (UH) and Vlaamse Interuniversitaire Raad (VLIR-UOS) DESAFIO Program.

**Competing interests:** He authors have declared that no competing interests exist.

months later after the first outbreak, another outbreak emerged in Yambuku, Zaire, affecting 315 and killing 254 people [3].

In March 2014, a major Ebola outbreak emerged in Guinea, being declared by the World Health Organization as the largest ever documented outbreak in western African region. Later, the disease propagated to Liberia, Sierra Leone and Nigeria, and as of May it was reported that 27,443 people contracted the disease and about 11,207 died from EVD [4]. Furthermore, it is also reported that among all countries, Sierra Leone was the worst affected country, with 13,534 confirmed, probable, and suspected cases, as of 23rd May 2014 [5]. In light of this scenario, the government of Sierra Leone declared a State of Emergency on 6 August 2014 which was followed by 3 day nationwide quarantine on 19-21 September 2014 [6]. The EVD not only forced restricted measures in Sierra Leone, but also urged the World Health Organization to declare the Ebola outbreak as an international public health problem of extreme emergency, turning it into a worldwide problem [4, 7].

The recurrent occurrence of Ebola outbreaks has boosted up the interest in understanding the dynamic transmission of the disease, and since its first outbreak in 1976 many studies have been carried out. For instance, Chowell et al. [8] analysed the Congo and Ebola outbreak data using a deterministic susceptible-exposed-infectious-removed (SEIR) model, coming up with deterministic ordinary differential equations for all the model parameters taking into account the effects of public health measures. Althaus [9] studied the Ebola outbreak data from Liberia, Guinea, and Sierra Leone, using a similar approach as [8], giving country-specific estimates. Both Chowell et al. [8] and Althaus [9] assume that the transmission rate is constant in the early days of the epidemic and decays exponentially with intervention measures. Nishiura and Chowell [10] and Chowell and Nishiura [11] give a review of epidemic modeling performed during the 2014 outbreak in Sierra Leone, Guinea and Liberia. Santermans et al. [12] present an extension of the SEIR model which incorporates disease-related mortality by making the distinction between survivors and non-survivors. Recently, Frasso and Lambert [13] analysed Ebola outbreak from Sierra Leone using an almost identical approach as Santermans et al. [12], though they use penalized B-splines to model the transmission rate, and in addition to that, use the Bayesian inferential framework for estimation.

Despite the vast number of studies in the framework of modeling infectious disease, the vast majority of studies use deterministic models, mainly susceptible-infectious-removed (SIR) in which the transition rates between states are estimated based on a system of ordinary differential equations. Further, oftentimes, such models are fitted to cumulative cases, which are extremely inter-correlated violating the assumption of independence required from fitting methods such as ordinary least square or maximum likelihood [14]. King et al. [15] argue that such models should be avoided, as they tend to overestimate the predictions on the course of the epidemic and substantially underestimate the uncertainty around the estimates and the predictions. As a matter of fact, stochastic models which have room for randomness are receiving increasing attention in modeling infectious disease. Lekone and Finkenstädt [16] describe a stochastic SEIR-model, as a counterpart of the deterministic SEIR model proposed by Chowell et al. [8].

In this manuscript, we investigate the spatio-temporal heterogeneity among all districts of Sierra Leone during the 2014 Ebola outbreak. Similar to Lekone and Finkenstädt [16], we use a stochastic-discrete SEIR model. Existing methodology, oftentimes, when fitting stochastic-discrete-time SEIR model, either consider fitting the model at aggregate national level data or conduct separate analyses for aggregate county level data. The latter approach, when the number of counties is considerably large, can be cumbersome. In this article, we use the Bayesian hierarchical framework to investigate the heterogeneity in the transmission among districts to better understand and model the disease outbreak. Transmission rates are allowed to be time-varying.

The rest of the paper is organized as follows. In the methodology section we present the data used for this study and with a brief review of the approach by Lekone and Finkenstädt [16] and its translation into the Bayesian hierarchical framework. In the result section, we present the posterior estimates of key epidemiological parameters and also evaluate the temporal variation of the estimated effective reproductive number. Moreover, we assess the robustness of the model with respect to the prior distributions. We then end the manuscript with a discussion, including conclusions, and recommendations for further research.

## Materials and methods

### Data description

The data comprise of cumulative cases of EVD from 3 July 2014 to 27 June 2015 reported by the Sierra Leone Ministry of Health and Sanitation [5]. Throughout the aforementioned period, the country reported a total of 12,790 Ebola cases, which is a summation of all confirmed, probable, and suspected cases. The weekly number of cases in Sierra Leone is presented in Fig 1A. Over the course of the epidemic, the country registered its highest peak in October 2014 reporting a total of 865 cases in a week. After 23 October 2014, the number of new cases started decreasing, and another peak was observed in April 2015. Fig 1B shows the final size of the outbreaks in the different districts of Sierra Leone. Among all district, Western Urban, Port Loko, and Western Rural were the most affected by the epidemic, reporting a total of 3459, 2108, and 1634, respectively, while Pujehun and Bonthe were the least affected, reporting a total of 89 cases each.

In the beginning of the epidemic, the data were recorded every two to three days, and later on a daily basis. As the data were irregularly reported throughout the outbreak period, for practicability and for reasons that will be clear later, we worked with weekly data. While the reported data are cumulative cases of EVD, we use incidence data. In mathematical perspective, both cumulative and incidence data convey the same information, which is not true in statistical perspective, as cumulative incidence data are correlated. Furthermore, fitting epidemic models to cumulative incidence data can lead to erroneous conclusions—bias, underestimation in the epidemiological quantities and forecasts [15, 17] As the reported cumulative cases are not monotonically increasing curves, we monotonized the cumulative cases using the PAVA algorithm [18], before computing the newly cases of EVD, to avoid the occurrence of negative newly cases of EVD.

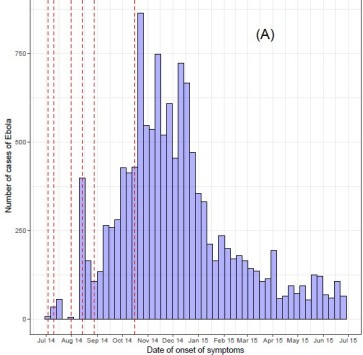
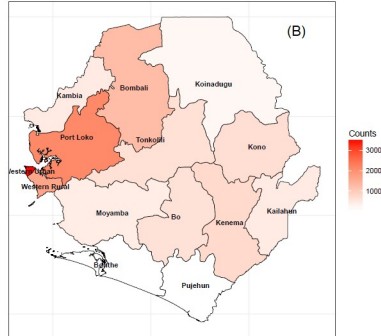

**Fig 1. Data from 2014 Ebola outbreak in Sierra Leone from July 3, 2014 to June 27, 2015.** (A) Daily counts of Ebola weekly counts. The red dotted vertical lines show the starting points of the epidemic in districts where the epidemics starts later. (B) Total number of Ebola cases by districts.

In some districts, the epidemic started later (see vertical lines in Fig 1A for the dates of first reported case). We assume that there were no cases in districts for weeks where there was no record of EVD.

## Transmission model

To study the dynamic transmission of EVD, we use a stochastic discrete-time SEIR-model proposed by Lekone and Finkenstädt [16] which is a counterpart of a deterministic SEIR-model used by Chowell et al. [8] and Althaus [9] as shown in Eq (1). The SEIR model assumes that the population is divided into 4 compartments: the susceptible compartment $S$, the exposed compartment $E$, the infected compartment $I$ and the removed compartment $R$ (which includes all individuals who either recover from the disease or die). The deterministic model is given by the following nonlinear ordinary differential equations ([8, 9, 19])

$$\dot{S}(t) = -\beta(t)S(t)I(t)/N$$
$$\dot{E}(t) = \beta(t)S(t)I(t)/N - \varrho E(t)$$
$$\dot{I}(t) = \varrho E(t) - \gamma I(t) \tag{1}$$
$$\dot{R}(t) = \gamma I(t),$$

where $S(t)$, $E(t)$, $I(t)$ and $R(t)$ denote the number of susceptible, exposed, infected and recovered individuals at time $t$, and $\dot{S}(t)$, $\dot{E}(t)$, $\dot{I}(t)$ and $\dot{R}(t)$ their corresponding derivatives with respect to time. Individuals in the susceptible compartment $S$ enter the exposed compartment $E$ at rate $\beta(t)I(t)/N$, with $\beta(t)$ the per capita transmission rate and $N$ the population size. Individuals in the exposed compartment $E$ move to the infectious compartment $I$ at a per capita rate $\varrho$ and from the infectious compartment enter the removed compartment $R$ at per capita rate $\gamma$.

The stochastic discrete-time SEIR-model relates directly to its deterministic counterpart Eq (1), but assumes that the transitions from one compartment to another are probabilistic. Let $B(t)$ record the number of susceptible individuals who become infected, $C(t)$ the number of cases by date of symptom onset, and $D(t)$ the number of cases who are removed (either die or recover from the disease) from the infectious compartment within the time interval $(t, t + h]$, where $h$ represents the difference between the time points at which data are collected. As the data were not collected on a daily basis, and to facilitate matters, $h$ is set to be equal to 1 week. The transitions from one compartment to another are shown in Fig 2. Consider that after being infected, individuals stay in the exposed population for a random duration $T_1$, and then they are infectious for a random duration $T_2$ before leaving the infectious population. Similar to Lekone and Finkenstädt [16], it is assumed that both $T_1$ and $T_2$ are independent and identically distributed according to an exponential distribution with rate $\varrho$ and $\gamma$, respectively. During their infectious period, they expose new individuals according to an in-homogeneous

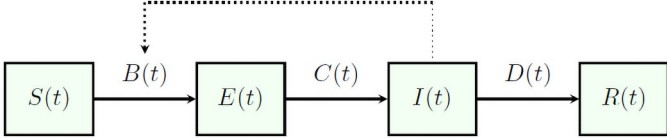

**Fig 2. Representation of SEIR-model.** Dynamics of the susceptible ($S(t)$), exposed ($E(t)$), infectious ($I(t)$) and recovered ($R(t)$) individuals. Solid arrows denote transitions between states. Dotted arrow indicates that transmission depends on the interactions between ($S(t)$) and ($I(t)$).

Poisson process with intensity $\beta(t)\frac{I(t)}{N}$. As a result, the transition probability from one state to another is given by $1 - \exp(-\lambda(t)h)$, where $\lambda(t)$ denotes the transition rate $\frac{\beta(t)I(t)}{N}$, $\varrho$ and $\gamma$ for susceptible to exposed, exposed to infectious, and infectious to removed compartment, respectively. This discrete-time stochastic model can be written as the following set of difference equations ([16]):

$$S(t + h) = S(t) - B(t)$$

$$E(t + h) = E(t) + B(t) - C(t)$$

$$I(t + h) = I(t) + C(t) - D(t) \tag{2}$$

$$S(t + h) + E(t + h) + I(t + h) + R(t + h) = N$$

with initial condition $S(0) = s_0$, $E(0) = e_0$ and $I(0) = i_0$. The transition from one state to the other is modeled as

$$B(t) \sim \mathrm{Bin}(S(t), P(t)), \quad C(t) \sim \mathrm{Bin}(E(t), p_C), \quad D(t) \sim \mathrm{Bin}(I(t), p_R) \tag{3}$$

with

$$P(t) = 1 - \exp\left[-\frac{\beta(t)I(t)}{N}h\right], \quad p_C = 1 - \exp(-\varrho h), \quad p_R = 1 - \exp(-\gamma h). \tag{4}$$

which resembles a discrete-time-Markov-chain process.

From this model, one can see that the transitions from one state to another are governed by the binomial distributions in Eq (3). Lekone and Finkenstädt [16] assume that the time-varying transmission rate $\beta(t)$ is constant until the point when control measures are introduced and thereafter decreases exponentially. This is a very popular assumption in the literature of susceptible-infectious-removed (SIR) and SEIR-model. However, this assumption seems to be rather restrictive as not only intervention measures play a role in reducing the appearance of new exposures. As an alternative, we propose to model the transmission rate $\beta(t)$ as a step function, i.e,

$$\beta(t) = \sum_{k=1}^{K} \beta_k \mathbb{1}_{A_k}(t), \tag{5}$$

where $K > 0$, $\beta_k$ are transmission rates over some intervals $A_k$, and $\mathbb{1}_A$ is the indicator function of $A$. We assume that the interval $A_k$ spans a period of one month, that is, for each month the transmission rate is constant. Alternatively, instead of taking equally fixed spaced periods of 30 days, one could estimate the number of change points, i.e, the points at which the transmission rate changes. Nevertheless, such formulation brings more complexity into the model [20].

Another assumption of the model in Lekone and Finkenstädt [16] is that the population mixed homogeneously (each individual is equally likely to contract the disease). This assumption is very common in SIR and SEIR models. Nevertheless, this assumptions is known to be unrealistic as mixing in the population depends on many factors, such as age (children usually have more effective contacts than adults), susceptibility to disease, position in space and the activities and behaviors of individuals [21, 22]. In order to relax this assumption, we will investigate a stochastic discrete-time multi-population version of Eqs (2) and (3). Let $S_j(t)$, $E_j(t)$, $I_j(t)$ and $R_j(t)$ denote the total number of susceptible, exposed, infectious, and recovered in district $j$, respectively ($j = 1, 2, \ldots, n$). Similarly $B_j(t)$, $C_j(t)$ and $D_j(t)$ denote the number of susceptible individuals that become infected, the number that become cases and the number of cases who

are removed in district $j$, respectively. The stochastic multi-population SEIR model is given by the following set of difference equations:

$$S_j(t+h) = S_j(t) - B_j(t)$$

$$E_j(t+h) = E_j(t) + B_j(t) - C_j(t)$$

$$I_j(t+h) = I_j(t) + C_j(t) - D_j(t) \tag{6}$$

$$S_j(t+h) + E_j(t+h) + I_j(t+h) + R_j(t+h) = N_j.$$

Here it is also assumed that in each district the movements from one state to another are governed by the binomial distribution given in Eq (3), but now with district-specific parameters $\beta_j(t)$, $\varrho_j$ and $\gamma_j$. Likewise, as in the single population discrete time stochastic model, interest lies in estimating the district-specific transmission rates $\beta_j(t) = \sum_{k=1}^{K} \beta_{jk} \mathbb{1}_{A_k}(t)$, incubation periods $(1/\varrho_j)$, and infectious periods $(1/\gamma_j)$.

We make use of a hierarchical (random-effects) approach for the time- and district-specific parameters of interest. This not only helps to borrow information from neighboring time periods or regions, it also helps in avoiding the over-fitting due to numerous parameters [23]. For the district-specific transmission rates $\beta_j(t)$, it is assumed that $\beta_{jk}$ also have district-specific mean and variability, that is,

$$\beta_{jk} \sim \mathcal{N}(\mu_{\beta_j}, \sigma_{\beta_j}^2).$$

Note that in this formulation, we allow exchange of information overtime, rather than geographically. Likewise, independent normal random effects for the district-specific incubation period and infectious period are chosen:

$$1/\varrho_j \sim \mathcal{N}(\mu_{1/\varrho}, \sigma_{1/\varrho}^2)$$

and

$$1/\gamma_j \sim \mathcal{N}(\mu_{1/\gamma}, \sigma_{1/\gamma}^2).$$

One should keep in mind that, contrary to the district-specific transmission rates, where the smoothing is done overtime, for the incubation period and infectious period the smoothing is done spatially. This is a very flexible model, allowing all parameters to be district-specific. We will investigate whether or not there is heterogeneity in these parameters across districts, or whether it is plausible to assume that these parameters are constant across districts. Therefore, two additional models are considered. First, we simplify the model by assuming that the incubation period and infectious period are no longer district-specific but common across the districts. Further, the model is modified by assuming that the district-specific transmission rates not only exchange information temporally but also geographically, that is, $\beta_{jk} \sim \mathcal{N}(\mu_\beta, \sigma_\beta^2)$. The former and the latter are termed as Model 2 and Model 3, respectively.

## The effective reproductive number

Of major interest is the basic reproduction number generally denoted by $R_0$. It is defined as the average number of new cases generated by a single infectious individual throughout her/his infectious period when introduced into a totally susceptible population [24]. The basic reproduction number ($R_0$) is of vital importance to assess the future outcome of the epidemic —if $R_0 \leq 1$, the epidemic will die out, and if $R_0 > 1$, the epidemic will take off. However, the

basic reproduction number, as defined by Diekmann et al. [24], only applies to the beginning of the epidemic.

Therefore, we rather consider the effective reproductive number, $R_t$, defined as the expected number of secondary cases per primary case over the course of the epidemic [10]. The effective reproduction number is of extreme importance as it accounts for external influences on transmission such as variations in human behavior and introduction of intervention measures. For model Eqs (1) and (2), Lekone and Finkenstädt [16] and Chowell et al. [8] define the effective reproduction number as follows:

$$R_t = \frac{\beta(t)}{\gamma} \frac{S(t)}{N}. \tag{7}$$

If one considers that the size of the outbreak is negligible as compared to the total population size, i.e, $S(t) \approx N$, the effective reproduction number can be approximated by $R_t \approx \beta(t)/\gamma$. This approximation is also valid for model Eq (6).

## Inference and model building

The multi-population discrete time stochastic model Eq (6) is fitted to weekly incidence data. From this model, only the time series $C(t)$ (the number of cases by date of symptom onset) is observable. The absence of information on exposure, as well as on recovery hinders the estimation process through maximum likelihood. Therefore, the Bayesian paradigm implemented through MCMC methods is implemented. This approach allows one to numerically integrate over the probability distribution of the unobserved processes [16, 25]. The unobserved data on exposure $B(t)$ and on recovery $D(t)$ are imputed using the conventional Metropolis-Hastings algorithm [26]. Moreover, the Bayesian framework paradigm allows to encompass parameter uncertainty and incorporate prior information in the estimates of transmission rate, incubation period, and infectious period.

The formulation of the Bayesian paradigm depends on the elicitation of the prior distributions on the parameters. Because of limited information in incidence data about the infectious period and incubation period, and the unobserved components in the SEIR model, informative priors will have to be used on these parameters to enhance valid model estimation. Therefore, informative hyper-priors on the mean transmission rate $\mu_{\beta j}$, mean incubation period $\mu_{1/\varrho}$, and mean infectious period $\mu_{1/\gamma}$ are chosen based on previous studies on Ebola. The following priors were chosen $\mu_{\beta j} \sim \mathcal{N}(1.4, 0.00142)$, $\mu_{1/\varrho} \sim \mathcal{N}(1.29, 0.007)$, $\mu_{1/\gamma} \sim \mathcal{N}(1, 0.00064)$. The variance parameters are given an inverse gamma prior, IG(100, 1). These elicited priors are in line with other studies on Ebola outbreaks [13, 16, 27, 28]. Furthermore, for Model 2, it is no longer assigned an hyper-prior to the distribution of incubation period and infectious period but instead it is considered that $1/\varrho \sim \mathcal{N}(1.29, 0.007)$ and $1/\gamma \sim \mathcal{N}(1, 0.00064)$, and for Model 3 it is assumed that $\mu_\beta \sim \mathcal{N}(1.4, 0.00142)$, with the same priors on the incubation period and infectious period as Model 2. Note that, to address identifiability issues, the elicited prior distribution for the mean transmission rate and the infectious period are tighter than those specified by Lekone and Finkenstädtand [16], and Frasso and Lambert [13]. The initial numbers for the model were taken as follows: $S_j(0) = N_j - E_j(0) - I_j(0)$. To find compatible values of $E_j(0)$ and $I_j(0)$ with the observed incidence data, uniform vague priors were assigned to the number of exposed and infected individuals at the start of the epidemic, $E_j(0) \sim U(0, 1000)$ and $I_j(0) \sim U(0, 20)$, and thus, allowing them to be estimated from the data.

To evaluate the joint posterior distribution, a chain of 500$k$ iterations with thinning of 100 was considered. Posterior estimates were computed after discarding a burn-in of 100$k$ iterations. Adequacy of model fit was assessed by means of the Deviance Information Criterion

(DIC) [29], where a lower DIC provides a good balance between model fit and complexity. Moreover, stochastic simulations were used to assess the performance of the selected best candidate model, by means of DIC. Evaluation of the posterior distribution was done using JAGS version 4.3.0. JAGS was run using the package jagsUI [30] from R, a statistical computing software [31]. The code used to evaluate the posterior distribution is available at https://github.com/RachidMuleia/Sierra_Leone_Data/blob/master/SEIR_epidemic_model_code.

## Results

### District specific posterior estimates

Three scenarios were considered. The first scenario is the most flexible, allowing all model parameters to vary across the districts (incubation period, infectious period, mean transmission rate and variance of the transmission rate). In the second scenario, the model is gradually simplified model, assuming incubation period and infectious period common across the districts. Then, the model is further simplified by considering that the mean transmission rate is not district-specific but common, allowing the districts to exchange information in the estimation of the transmission rate. Among all the models, Model 3 outperforms all the other models, reporting the lowest DIC 7290.034. Next follows Model 2, with the second lowest DIC 8284.402. Model 1, considered being the most flexible, based on DIC, appears to result in the worst fit DIC = 8977.134. Hence, we see that it is more plausible to assume a common incubation and infectious period. This is also confirmed by the agreement of the posterior estimates under Model 1 and Model 2 (see S1 and S2 Tables). Therefore, inference follows considering Model 3. Additional discussion on the plausibility of Model 3 is presented in the Appendix.

Table 1 shows the posterior summary statistics for mean transmission rate ($\mu_\beta$), incubation period ($1/\varrho$), infectious period ($1/\gamma$), and the variance of the transmission rate ($\sigma_\beta^2$). From this table, it is observed that the posterior mean transmission rate is estimated at 1.21 weeks$^{-1}$ (95% CI: 1.14—1.28). It is also observed that the incubation period is estimated to be 2.30 weeks (95% CI: 2.20–2.43) and the infectious period is estimated at 1.07 weeks (95% CI: 1.03—1.11). It is worth emphasizing that the posterior estimate of the mean transmission rate and the infectious period are a reflection of the information contained in the prior distribution, as the available incidence data contain only limited information about these parameters. This is exactly why these parameters are informed by the information from literature.

The Bayesian inference also allowed us to estimate the effective reproductive number ($R_t$) at each iteration. The posterior temporal variation of $R_t$, for each of the 14 districts, is displayed in Fig 3. It can be seen that the reproductive number ranges from below 1 to 5.2. Additionally, it is apparent that the effective reproductive number is consistent with the data as illustrated in Fig 3. Nevertheless, it can be seen that in districts with less data, the posterior effective

**Table 1. Posterior summary statistics for Model 3 on district specific data.**

| Parameters | Mean | SD | Quantiles | | |
| --- | --- | --- | --- | --- | --- |
| | | | 2.5% | 50% | 97.5% |
| Mean transmission rate ($\mu_\beta$) | 1.21 | 0.04 | 1.14 | 1.21 | 1.28 |
| Incubation period ($1/\varrho$) | 2.30 | 0.05 | 2.20 | 2.30 | 2.40 |
| Infectious period ($1/\gamma$) | 1.07 | 0.02 | 1.03 | 1.07 | 1.11 |
| Transmission rate variance ($\sigma_\beta^2$) | 0.42 | 0.04 | 0.34 | 0.41 | 0.51 |

The posterior summary statistics are computed from a chain of 500$k$ iterations after a burn-in of 100$k$ iterations. This model assumes that the mean transmission rate, the incubation period and the infectious period are common for all the districts.

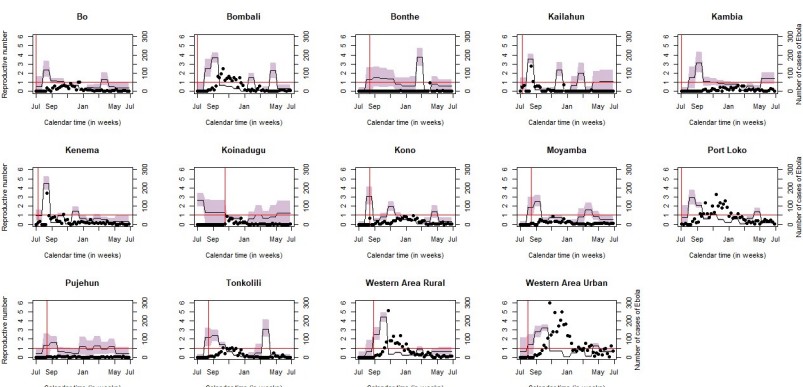

**Fig 3. Effective reproductive number (black solid line) with its 95% credible interval.** The horizontal dashed red line corresponds to $R_t = 1$. The vertical dashed red line indicates where exactly the epidemic starts. These results correspond to the model with informative priors. The curves were produced based on Model 3. The model assumes constant mean transmission rate, constant incubation and constant infectious period across the districts.

reproductive number shows substantial uncertainty. We observe that in Bo, Bonthe, Pujehum and Moyamba the effective reproductive is below 1.0, almost through the entire period. This suggests that in these districts there was no major Ebola outbreak, that is, the outbreak could not sustain itself. In the districts Bombali, Western Area Rural, Western Area Urban and Port Loko, we observe, in the early weeks of the epidemic, an increase of the effective reproductive number, and sometimes a decrease is observed in December. Furthermore, around April and July 2015, in almost all districts (Bo, Bombali, Bonthe, Kono, Moyamba, Tonkolili and Western Area Urban) we observe a rise in the effective reproductive number, a pattern that accompanies the data. The rise of the effective reproductive number in the late weeks of the epidemic could be associated to resurgences of new cases of EVD.

## Assessing the fit

To assess the fit of our model, we do not only use the DIC but we also perform a stochastic simulation using the obtained posterior distribution. The fit is assessed by comparing the simulated incidence data with the observed curves of EVD. The simulated incidence data are obtained by first randomly sampling 100 vectors of the marginal posterior distribution, $(\beta_j(t)$, $1/\varrho$, $1/\gamma$, $E(0)$, $I(0))$, and then simulate one epidemic for each sampled vector. The simulated curves together with the observed incidence data, for the district-specific model, are displayed in Fig 4. To have a clear depiction of the simulated data and the observed incidence data, they are plotted in logarithm scale—$\log_{10}(x + 1)$. From this figure, it can be observed that the model manages to predict the course of the epidemic very well, with some digressing patterns for districts with few cases of EVD (Bonthe, Koinadugu, Kambia and Pujehun). This suggests that the multi-population SEIR model does not provide a good fit for districts with sporadic cases of EVD. In fact, Getz et al. [32] studied the adequacy of SEIR model for the EVD outrbeak in Sierra Leone when the epidemic has a spatial structure and noted that the model fails to provide a good fit for districts with few cases of EVD, and as a remedy, they opt to ignore districts with few or sporadic cases of EVD.

## Nationwide posterior estimates

Besides the district-specific analysis, we have also conducted a nationwide analysis. This analysis is performed considering the least complex model, that is, Model 3. Similar prior

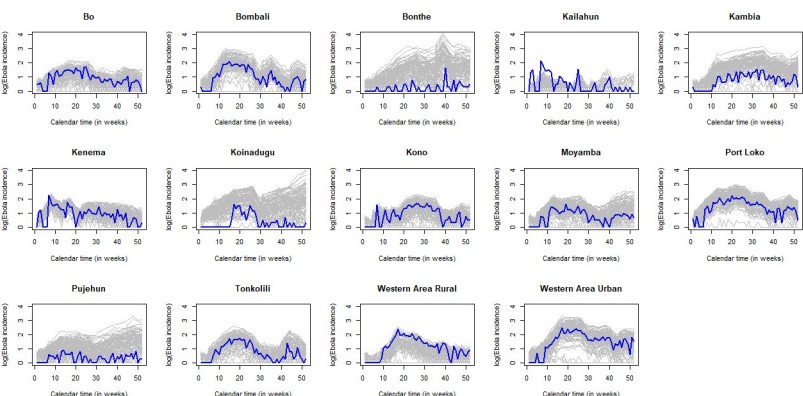

**Fig 4. Stochastic simulation of the incidence data obtained from the joint posterior distributions of ($\beta_j(t)$, $1/\varrho$, $1/\gamma$, $E(0)$, $I(0)$).** The simulated data and the observed incidence data are displayed on the logarithm scale. The solid blue curve represents the observed incidence data and the solid gray curves represent the simulated incidence data.

distributions as in the district-specific scenario have been assumed. Posterior estimates of $\mu_\beta$, $1/\varrho$, $1/\gamma$, and $\sigma_\beta^2$ are reported in Table 2. It can be seen that $\mu_\beta$ is estimated at 1.22 weeks$^{-1}$. The incubation period and the infectious period are estimated at 1.66 weeks and 1.08 weeks, respectively. It can be clearly noted that the posterior estimates of these parameters are approximately equal to the means of their respective prior distributions. This is due to the fact that only limited information is contained in the incidence data about the mean transmission rate and the infectious period. Moreover, it can be observed that the posterior estimates of the mean transmission rate and the infectious period of the nationwide model are very similar to their counterparts in the district specific analysis. This is expected, as the information on these parameters come predominantly from the elicited prior distribution. With regards to the incubation period, it is observed that the model on nationwide data provides smaller posterior estimates.

The nationwide analysis also allowed us to compute the posterior curve for the effective reproductive number. Fig 5A shows the posterior estimate of $R_t$. It can be observed that the reproductive number, in the entire outbreak period, varies from below 1.0 to 2.5. In the district-specific analysis, it was noted that the interval was much wider as compared to the nationwide analysis. This could be attributed to the fact that, in the district-specific analysis, the data is rather sparse. Moreover, the curve of the effective reproductive number illustrates a reproductive number greater than one before November. This result was also observed in most of the districts in the district-specific analysis, mainly in districts with larger outbreak. The graph further reveals that the pattern of the effective reproductive number is consistent with the

**Table 2. Posterior summary statistics for Model 3 on nationwide data.**

| Parameters | Mean | SD | Quantiles | | |
| --- | --- | --- | --- | --- | --- |
| | | | 2.5% | 50% | 97.5% |
| Mean transmission rate ($\mu_\beta$) | 1.22 | 0.04 | 1.15 | 1.22 | 1.30 |
| Incubation period ($1/\varrho$) | 1.66 | 0.07 | 1.53 | 1.66 | 1.79 |
| Infectious period ($1/\gamma$) | 1.08 | 0.02 | 1.03 | 1.08 | 1.12 |
| Transmission rate variance ($\sigma_\beta^2$) | 0.0403 | 0.0050 | 0.0313 | 0.0400 | 0.0512 |

The posterior summary statistics are computed from a chain of 500$k$ iterations after a burn-in of 100$k$ iterations.

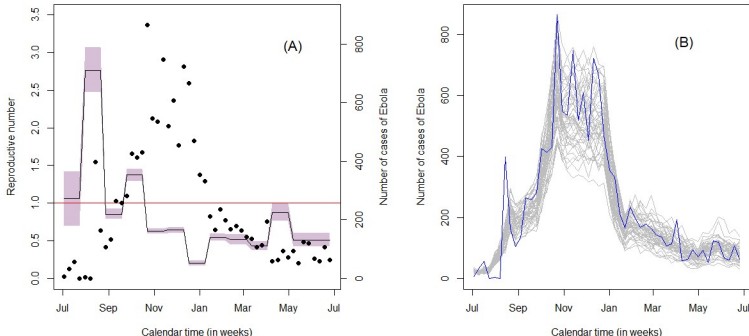

**Fig 5. Effective reproductive number and simulated incidence of EVD.** (A) Posterior effective reproductive number (black solid line) for the nationwide data with its 95% credible interval(shaded area). The horizontal red line corresponds to the epidemic threshold, $R_t = 1$. The black dots correspond to the observed cases of Ebola. (B) Stochastic simulation of the incidence data from the marginal posterior distribution of $(\beta(t), 1/\varrho, 1/\gamma, E(0), I(0))$.

observed data. As a matter of fact, the epidemic starts declining around December, hence it can be observed that, from December onward, the effective reproductive number is below one. This suggests that the models nicely describes the epidemic. As in the district-specific scenario, we also assess the fit of the model on the nationwide data by running stochastic simulations. The simulated incidence data together with the observed Ebola incidence data are presented in Fig 5B. The simulated epidemic curves are consistent with the observed data, suggesting that the parameters are reasonably well estimated.

## Sensitivity analyses

To assess the robustness of our estimates with respect to the specification of prior distribution, a sensitivity analysis is conducted on Model 3 and on the nationwide model. Weakly informative priors are assigned on the $\mu_\beta$, $1/\varrho$ and $1/\gamma$, by slightly increasing the variance of the previously specified prior; that is, $\mu_\beta \sim \mathcal{N}(1.40, 0.0142)$, $\mu_{1/\varrho} \sim \mathcal{N}(1.29, 0.07)$ and $\mu_{1/\gamma} \sim \mathcal{N}(1, 0.0064)$. The results from these analyses are shown in Table 3, and in S3 and S4 Figs. From the table, a considerable change in the posterior parameter estimates for both district-specific and nationwide model is observed. The results further suggest that a "loose" prior distribution leads to a larger incubation period and larger infectious period. It can also be noted that the posterior estimates for the district-specific model and the nationwide model are quite similar, except for the variance of the transmission rate. Additionally, it is noted that there are no substantial changes in the patterns of the effective reproductive number. However, considerable uncertainty is observed, mainly in districts with few cases of EVD. Furthermore, it is seen that posterior estimates of the effective reproductive number, under weakly informative priors, are slightly larger than those under informative priors. The larger amount of uncertainty observed under weakly informative priors is expected because of the unobserved components in the SEIR model (see S3 and S4 Figs). Note that the effective reproductive number obtained from the nationwide model does not show substantial uncertainty to the extent as that obtained from the district specific model. This suggests that sufficient amount of data is necessary for producing precise estimates.

We also investigated the plausibility of the assumption of a time dependent transmission rate. To test this assumption, we assess the performance of our model when constant transmission rate is assumed, limiting the analysis to the nationwide data. The assumption of constant

**Table 3. Posterior summary statistics for Model 3.**

**District-specific model**

| | | | Quantiles | | |
| --- | --- | --- | --- | --- | --- |
| Parameters | Mean | SD | 2.5% | 50% | 97.5% |
| Mean transmission rate ($\mu_\beta$) | 0.71 | 0.06 | 0.59 | 0.71 | 0.83 |
| Incubation period ($1/\varrho$) | 2.70 | 0.34 | 2.04 | 2.85 | 3.08 |
| Infectious period ($1/\gamma$) | 1.59 | 0.11 | 1.40 | 1.58 | 1.81 |
| Transmission rate variance ($\sigma_\beta^2$) | 0.24 | 0.04 | 0.16 | 0.24 | 0.31 |

**Nationwide model**

| | | | Quantiles | | |
| --- | --- | --- | --- | --- | --- |
| Parameters | Mean | SD | 2.5% | 50% | 97.5% |
| Mean transmission rate ($\mu_\beta$) | 0.74 | 0.05 | 0.65 | 0.74 | 0.84 |
| Incubation period ($1/\varrho$) | 2.14 | 0.14 | 1.88 | 2.14 | 2.41 |
| Infectious period ($1/\gamma$) | 1.49 | 0.06 | 1.37 | 1.49 | 1.62 |
| Transmission rate variance ($\sigma_\beta^2$) | 0.026 | 0.003 | 0.020 | 0.026 | 0.033 |

The results in this table correspond to the posterior summary statistics for Model 3, which is fitted to district-specific and nationwide data, considering weakly informative priors. The posterior summary statistic are computed from a sample of 500$k$ iteration after a burn-in of 100$k$ iterations.

transmission rate applies only to the beginning of the epidemic (see Ganyani et al. [33]). Therefore, the model with constant transmission rate is fitted to the EVD outbreak data considering only the first 17 weeks of the epidemic (at week 17 the outbreak reaches its peak). When a constant transmission rate is assumed, we notice that $R_0$ is estimated as 1.33. One can also notice that the incubation period is slightly larger as compared to the incubation period under the assumption of a time-dependent transmission rate. Regarding the infectious period, the posterior estimate does not vary considerably. In fact, the posterior estimate for the infectious period and for the transmission rate reflect the information coming from their prior distributions (see S3 Table). Although the estimates of the infectious period and the incubation period only vary marginally, taking the transmission rate to be constant is not a plausible assumption. As it can be seen from S5 Fig, assuming constant transmission rate substantially under-estimates the predictions.

## Discussion

In this article, we make use of a stochastic multi-population SEIR model, an extension of the discrete-time stochastic model proposed by Lekone and Finkenstädt [16], to analyze the 2014 Ebola outbreak in all districts of Sierra Leone. In the model we account for heterogeneity amongst district through a hierarchical model for the district-specific parameters, which allows borrowing of information across neighboring regions [23]. Additionally, as local control measures and other unmeasured factors might play a role in the transmission process, district-specific time-varying transmission rates are considered. Other functional form such as P-splines as been used by Frasso and Lambert [13] or Wiener processes as in Camacho et al. [34] could have been used.

The proposed model was fitted to the weekly incidence data with the main purpose of taking the heterogeneity into account when estimating the transition parameters between compartments, together with the effective reproductive number. Likewise the model used by Lekone and Finkenstädt [16], our model lacks information on exposure and recovery. The

absence of this information prompted us to use Bayesian formalism for parameter estimation, with the incorporation of informative priors. Nevertheless, this induces an additional source of variability, and this was confirmed by wider credible interval of the effective reproductive number $R_t$. In the estimation process three scenario were considered: a flexible model allowing all parameters to be districts specific, a model with constant incubation period and constant infectious period, and a model with constant mean transmission rate, constant incubation and constant infectious period. The results showed that the latter scenario—the one with constant mean transmission rate, constant incubation period and constant infectious period—is the most plausible. In fact, this was expected as incubation and infectious period are disease specific characteristic. Moreover, we have assessed the fit of our model through stochastic simulations. The simulations revealed that our model (with common mean transmission rate and constant incubation period and constant infectious period) appropriately describes the epidemic both at district and nationwide level.

The results showed no common pattern of the effective reproductive number across the districts, indicating that intensity of transmission is highly variable across the districts. The estimated reproduction numbers at district level ranged from below 1.0 to 4.5. These results are consistent with a study by Kucharski et al. [35]. At nationwide level, the reproduction number in the early stages of the epidemic was estimated to range from 1.0 to 2.9. These findings are in line with Nishiura and Chowell [10]. The authors analysed the Sierra Leone Ebola incidence data in the early stages of the epidemic and found reproduction numbers between 1.4 and 1.7. Similarly, Althaus [9], using an ODE with a time-dependent transmission rate, estimated the reproduction number for Sierra Leone to range between 1.0 and 2.5, in the early months of the epidemic. Gomes et al. [36], using a multimodel inference approach and accounting for international spread, estimated the reproduction number for the Ebola outbreak in West Africa to range from 1.5 to 2.5. Fisman et al. [37] found an overall estimate of the basic reproduction number between 1.4 and 2.6 for the West Africa Epidemic. Nevertheless, for Sierra Leone in particular, they probably overestimated the basic reproduction number at 8.33. Using viral genetic sequencing data, Stadler et al. [38] reported a median estimate of $R_0$ between 1.65 and 2.18. The WHO Ebola Response Team [27] also analysed the Ebola incidence data in three most affected countries—Guinea, Liberia and Sierra Leone. They estimated for Sierra Leone an $R_0$ of 2.02 (1.79-2.26), which is in accordance with our estimates of $R_t$ at district and nationwide level. The estimates of $R_t$ herein are also consistent with previous small outbreaks in Sudan and Zaire. The estimate of $R_0$ for the Ebola outbreak in Zaire varied between 1.4 and 4.7, and in Sudan the estimate ranged from 1.3 to 2.7 [39]. Furthermore, we were able to conclude that $R_t$ started decreasing around December to July 2015. Frasso and Lambert [13] also concluded that the $R_t$ approached 1.0 by the end of December 2014. Districts like Bo, Bonthe, Kambia, Koinadugu and Pujehum showed a fluctuation of $R_t$ around the threshold overtime, suggesting that the epidemic is not sustainable. The fluctuation around the threshold could be an indication of the effectiveness of the control measures in these districts. These results tie well with what was found by Santermans et al. [12] and Camacho et al. [34]. Similar to Santermans et al. [12] and Camacho et al. [34], we observed a substantial heterogeneity in the estimated effective reproductive number, temporally and geographically. The observed heterogeneity in the estimated $R_t$ could be attributed to differences in the size of the susceptible population among districts as well as to cultural and behavioral differences. Furthermore, the differences in the transmission of Ebola may be justified by the varying interventions implemented in each districts.

In the present study, the multi-population model also allowed us to estimate the incubation period and the infectious period. Using the the district-specific transmission model, with common incubation period and common infectious period, the mean incubation period was

estimated at 2.30 weeks (95% CI: 2.20-2.40). Our estimate for the mean incubation period, although it is within the minimum and maximum value for incubation period (2 to 21 days), is considerably larger when compared with what is reported in the literature. The nationwide model resulted into an estimate for the incubation period, 1.66 weeks $\sim$ 11.62 days (95%CI: 1.53-1.79), more in line with previous estimates of the incubation period. As a matter of fact, Okware et al. [40], for the 2000 Ebola outbreak in Uganda, reported an average incubation period of 12 days. A similar estimate was also found by Eichener et al. [28] for the 1995 Ebola outbreak in Kikwit, Democratic Republic of Congo. The authors estimated the mean incubation period to be equal to 12.7 days with a standard deviation of 4.31 days. Lekone and Finkenstädt [16], for the same Ebola outbreak, estimated an incubation period of 10.11 days. More recently, using retrospective data on exposure, the WHO Ebola Response Team [27] estimated an overall incubation period of 11.4 days, which was also found to be similar across the affected countries, Guinea, Liberia and Sierra Leone. The infectious period was estimated at 1.08 weeks (95% CI:1.03-1.12). Our estimate is consistent with other estimates for previous EVD outbreaks [8, 16]. A recent study on the latest West Africa EVD outbreak, using viral sequence data for different models, estimated an infectious period ranging between 1.22 and 6.09 days [38]. On the other hand, Frasso and Lambert [13] reported two estimates for the infectious period, one corresponding to the time from 1st symptoms to death (8.55 days) and another from the 1st symptoms to recovery (17.20 days). It should be recalled that the infectious period parameter is not identifiable, and the reported estimate only reflects the information contained in the respective prior distribution. Identifiability issues for the SEIR model parameters were also observed by Frasso and Lambert [13] and by Ganyani et al. [33].

The results further showed that the model is sensitive to prior specification, as the change in prior distributions impacted substantially the posterior estimates. Contrary to Lekone and Finkenstädt [16], weakly informative priors considered herein led to larger posterior estimates for the incubation period and infectious period. Moser et al. [41] studied the impact of prior information on the estimates of an epidemic model. They showed that slight modification can affect the posterior estimates. Additionally, they showed that using minimally informative prior leads to larger estimates of the $R_0$, a result that is concurrent with our findings. The sensitivity analyses, despite the larger values of the $R_t$ under weakly informative priors, showed that the pattern of the $R_t$ overtime is robust to prior elicitation. Moreover, the results revealed that, unlike the model with time-dependent transmission rate, the assumption of constant transmission rate leads to considerable discrepancy between the predicted incidence and the observed EVD incidence.

In general, the adopted Bayesian formalism appeared to be very flexible in the parameter estimation, mainly because it allowed us to provide inference even in the absence of information on exposure and recovery. It allows to relax the assumption of homogeneous mixing among the districts. The proposed model has some drawbacks. The model does account for heterogeneous mixing within the districts. For future work it could be interesting to consider this in the estimation process. Recently, Kong et al. [42] proposed a way of accounting for heterogeneity within the population by assuming that the number of contacts from individuals varies from person to person. Our model assumes perfect reporting of number of cases of EVD. Gamado et al. [43] show that ignoring underreporting leads to under-estimation of infection rate and reproductive number. Taking heterogeneous mixing and underreporting was beyond the scope of this research, but is an interesting extension for further research. The multi-population SEIR epidemic model could also be extended in several ways. For instance, Frasso and Lambert [13] use a SEIR epidemic model to analyse the Sierra Leone nationwide data. Nevertheless, in their approach they make distinction, in the removed compartment, between recovery and death. Such distinction is also possible for the multi-population SEIR

epidemic model. However, this could add complexity to the model and also result in identifiability issues, unless enough data are available to precisely estimate all the model parameters. Lekone and Finkenstädt [16] in their approach, to make use of death counts, unrealistically assume that individuals are only removed from the removed compartment through death, while in practice individuals can either be removed through death or recovery. Another extension that deserves an attention for future works is to consider transmission through contacts with dead bodies, as EVD can also be transmitted through contact with unburied dead bodies [44]. Santermans et al. [12] in their two stage modelling assessed the adequacy of their model when transmission through dead bodies is taken into account. Nevertheless, the fit of the model to the data did not improve, probably because no sufficient data were available to improve the model fit.

Despite the flexibility of Bayesian inferential techniques, lack of information leads to inaccurate estimates. Thus, we think that availability of exposure and recovery data would be insightful. Furthermore, availability of data on daily basis would also help in the quest of less uncertain posterior estimates. In fact, Lekone and Finkenstädt [16] noted that when data are collected on weekly basis, credible intervals were much wider.

## Supporting information

**S1 Appendix. Additional discussion and interpretation on Model 1 and 2.**
(PDF)

**S1 Table. Posterior summary measures for Model 1.** Posterior summary statistics for Model 1 computed from a sample of 500$k$ iterations after a burn-in of 100$k$. Model 1 considers that all parameter varies across districts. The model is fitted considering informative priors. 95% credible interval a reported in brackets.
(PDF)

**S2 Table. Posterior summary measures for Model 2.** Posterior summary statistics for Model 2 computed from a sample of 500$k$ iterations after a burn-in of 100$k$ iterations. Model 2 assumes common incubation period and common infectious period. The model is fitted considering informative priors. 95% credible interval a reported in brackets.
(PDF)

**S3 Table. Posterior summary measures for nationwide model with constant transmission rate.** Posterior summary statistic for for nationwide model with constant transmission rate. The model is fitted considering the first 17 weeks of the epidemic as the assumption of constant transmission rate applies only to the early stage of the epidemic.
(PDF)

**S1 Fig. Effective reproductive number for model 1 under informative priors.** Effective reproductive number(black solid line) with its 95% credible interval. The horizontal red line corresponds to $R_t = 1$. The vertical red line indicates where exactly the epidemic starts. These results correspond to the model with informative priors. The curves were produced based on Model 1. This model assumes that all the model parameters vary across the district. The black dots represent the observed incidence data.
(TIF)

**S2 Fig. Effective reproductive number model 2 under informative priors.** Effective reproductive number(black solid line) with its 95% credible interval. The horizontal red line corresponds to $R_t = 1$. The vertical red line indicates where exactly the epidemic starts. These results correspond to the model with informative priors. The curves were produced based on Model

2. This model assumes that incubation period and infectious period do not vary across the districts. The black dots represent the observed incidence data.
(TIF)

**S3 Fig. Effective reproductive number for model 3 under weakly informative prior.** Effective reproductive number(black solid line) with its 95% credible interval. The horizontal red line corresponds to $R_t = 1$. The vertical red line indicates where exactly the epidemic starts. These results correspond to the model with weakly informative priors. The model assumes constant mean transmission rate, constant incubation and constant infectious period. The curves were produced based on Model 3. The black dots represent the observed incidence data.
(TIF)

**S4 Fig. Effective reproductive number for nationwide model under weakly informative prior.** Effective reproductive number (black solid line) for the nationwide EVD data with its 95% credible interval. The horizontal red line corresponds to $R_t = 1$. These results correspond to the model with weakly informative priors. The black dots represent the observed incidence data.
(TIF)

**S5 Fig. Ebola incidence data versus simulated data from a stochastic epidemic model considering a constant transmission rate.** Stochastic simulation of the incidence data from the marginal posterior distribution of ($\beta$, $1/\varrho$, $1/\gamma$, $E(0)$, $I(0)$). The data are simulated for the first 17 weeks of the epidemic as the assumption of constant transmission rate applies for the early phase of the epidemic. The out break reaches the peak at week 17. The blue curve represents the observed EVD incidence data.
(TIF)

## Acknowledgments

The authors would like to express their gratitude to the editor and to the two anonymous reviewers for their valuable constructive comments and suggestions, which considerably improved the presentation of the paper.

## Author Contributions

**Conceptualization:** Rachid Muleia, Marc Aerts, Christel Faes.

**Data curation:** Rachid Muleia.

**Formal analysis:** Rachid Muleia, Christel Faes.

**Methodology:** Rachid Muleia, Christel Faes.

**Project administration:** Marc Aerts, Christel Faes.

**Software:** Rachid Muleia.

**Supervision:** Marc Aerts, Christel Faes.

**Visualization:** Rachid Muleia.

**Writing – original draft:** Rachid Muleia, Christel Faes.

**Writing – review & editing:** Marc Aerts, Christel Faes.

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
