## [Decision Letter · Decision Letter 0]

13 Nov 2020

PONE-D-20-21480

Multi-population Stochastic Modeling of Ebola in Sierra Leone: Investigation of Spatial Heterogeneity

PLOS ONE

Dear Dr. Muleia,

Thank you for submitting your manuscript to PLOS ONE. After careful consideration, we feel that it has merit but does not fully meet PLOS ONE’s publication criteria as it currently stands. 

Please carefully revise the manuscript, considering and addressing comments and questions raised by the two reviewers. They will be useful to improve the manuscript and make it feasible for publication in PLOS ONE. 

We look forward to receiving your revised manuscript.

Kind regards,

Maria Vittoria Barbarossa, Ph.D.

Academic Editor

PLOS ONE

Journal Requirements:

2.We note that [Figure(s) 1] in your submission contain map images which may be copyrighted. All PLOS content is published under the Creative Commons Attribution License (CC BY 4.0), which means that the manuscript, images, and Supporting Information files will be freely available online, and any third party is permitted to access, download, copy, distribute, and use these materials in any way, even commercially, with proper attribution. For these reasons, we cannot publish previously copyrighted maps or satellite images created using proprietary data, such as Google software (Google Maps, Street View, and Earth). For more information, see our copyright guidelines: http://journals.plos.org/plosone/s/licenses-and-copyright.

1.    You may seek permission from the original copyright holder of Figure(s) [1] to publish the content specifically under the CC BY 4.0 license. 

Reviewers' comments:

Reviewer's Responses to Questions

**Comments to the Author**

1. Is the manuscript technically sound, and do the data support the conclusions?

Reviewer #1: Partly

Reviewer #2: Yes

2. Has the statistical analysis been performed appropriately and rigorously? 

Reviewer #1: Yes

Reviewer #2: Yes

3. Have the authors made all data underlying the findings in their manuscript fully available?

Reviewer #1: Yes

Reviewer #2: Yes

4. Is the manuscript presented in an intelligible fashion and written in standard English?

Reviewer #1: Yes

Reviewer #2: Yes

5. Review Comments to the Author

Reviewer #1: Dear authors,

Please find my detailed review in the attached pdf file. Here, I simply copy my general assessment of your nice work.

The manuscript analyzes data from the Ebola outbreak in Sierra Leone in 2014 by a stochastic SEIR model to infer epidemic parameters by a hierarchical Bayesian model. For this, the authors study three statistical models: in the most complex model they allow for district-dependent incubation (transition E to I) and recovery (transition I to R) times, as well as for district- and time-dependent transmission rates (transition S to E); in the second complex model they study district- and time-dependent transmission rates and set the incubation and recovery time as district-independent; in the least complex model they additionally relax the assumption of district-specific transmission rates. Using the Deviance Information Criterion, the adapted version of the Akaike Information Criterion for hierarchical Bayesian models, the authors identify the least complex model as the most suitable one to explain the data. While this result is not surprising for the disease-specific parameters, i.e., the incubation and recovery time, it is interesting to see that assuming district-dependent transition rates does not provide a better fit to the data. Potential reasons for this observation could be discussed a bit more in my opinion.

In general, I think that the `Discussion' section is the weakest part of the manuscript. Briefly, I am missing that the authors put their results obtained from a stochastic district-specific SEIR model into context with results derived from other models and analyses. The authors do this for the effective reproductive number R_t but not for the estimated parameters. It would be interesting to see whether their detailed model provides new or other insights when compared to less detailed country-wide SEIR models. I would also suggest that they run their parameter inference on the country-wide data, i.e. accumulating all the data to a single data set, to see how the estimates compare. For more details on my concerns and suggestions related to the `Discussion' see below.

Overall, the model and its analysis are well-explained, and the analysis is, as far as I can tell, sound. I particularly liked the `Introduction' that nicely put this work into the context with other studies. Unfortunately, this was not the case in the `Discussion' which is the part that needs some major revisions in my opinion. I have also added a small list of minor comments below that the authors might want to take into account during their revision. Once my concerns regarding the `Discussion' are addressed, I would reconsider the manuscript for publication in PLOS ONE.

Reviewer #2: 1. The model parameters are allowed to vary in space (in one of the model variations), but they are assumed to be independent. what is the justification? IN the earth sciences, it is common to model spatial variability using a geostatistical model, so can you explain why you chose to ignore that option?

2. In modeling priors, I would imagine that you will try to compile information from previous studies, but to my understanding this was not done. Please explain.

3. I think the authors should benchmark their study against the previous studies. Specifically, how is their model working better than previous models?

4. in Line 48/49and 57 the authors state that ML cannot be used with correlated variables. This, to my understanding is inaccurate. You can take a look at publications from the earth sciences by authors such as P. Kitanidis and Y. Rubin (e.g., in Water Resources Research).

6. PLOS authors have the option to publish the peer review history of their article (what does this mean?). If published, this will include your full peer review and any attached files.

Reviewer #1: No

Reviewer #2: No

---

## [Author Response · Author response to Decision Letter 0]

14 Mar 2021

Reviewer 1: We have incorporated all the the suggestions and respond to all the comments. We conducted new analysis and do a thorough discussion on the paremeter estimates as recommended. Futher details are given on the pdf document

Reviewer 2: We have incorporated all the suggestion and comments from the your side. We also improved the discussion section as suggested by the reviewer. Details are given in the pdf document, with a point by point rebly.

---

## [Decision Letter · Decision Letter 1]

14 Apr 2021

Multi-population Stochastic Modeling of Ebola in Sierra Leone: Investigation of Spatial Heterogeneity

PONE-D-20-21480R1

Dear Dr. Muleia,

We’re pleased to inform you that your manuscript has been judged scientifically suitable for publication and will be formally accepted for publication once it meets all outstanding technical requirements.

Kind regards,

Maria Vittoria Barbarossa, Ph.D.

Academic Editor

PLOS ONE

Additional Editor Comments (optional):

Please address the minor comments by reviewer nr 1 before submitting the final version to the editorial office

Reviewers' comments:

Reviewer's Responses to Questions

**Comments to the Author**

1. If the authors have adequately addressed your comments raised in a previous round of review and you feel that this manuscript is now acceptable for publication, you may indicate that here to bypass the “Comments to the Author” section, enter your conflict of interest statement in the “Confidential to Editor” section, and submit your "Accept" recommendation.

Reviewer #1: All comments have been addressed

2. Is the manuscript technically sound, and do the data support the conclusions?

Reviewer #1: Yes

3. Has the statistical analysis been performed appropriately and rigorously? 

Reviewer #1: Yes

4. Have the authors made all data underlying the findings in their manuscript fully available?

Reviewer #1: No

5. Is the manuscript presented in an intelligible fashion and written in standard English?

Reviewer #1: Yes

6. Review Comments to the Author

Reviewer #1: The authors have done a great job in revising the manuscript and answering to my com-

ments. The results are much more convincing than before and put into context with previously

obtained estimates. I still have a few minor comments (see attached list), but apart from these

I believe that the manuscript is in a publishable state.

7. PLOS authors have the option to publish the peer review history of their article (what does this mean?). If published, this will include your full peer review and any attached files.

Reviewer #1: No

---

## [Editor Report · Acceptance letter]

30 Apr 2021

PONE-D-20-21480R1 

Multi-population stochastic modeling of Ebola in Sierra Leone: Investigation of spatial heterogeneity 

Dear Dr. Muleia:

I'm pleased to inform you that your manuscript has been deemed suitable for publication in PLOS ONE. Congratulations! Your manuscript is now with our production department. 

Kind regards, 

on behalf of

Dr. Maria Vittoria Barbarossa 

Academic Editor

PLOS ONE